# An End-to-End Atrous Spatial Pyramid Pooling and Skip-Connections Generative Adversarial Segmentation Network for Building Extraction from High-Resolution Aerial Images

**Mingyang Yu** [1][iD]**, Wenzhuo Zhang** [1,]*****, **Xiaoxian Chen** [1][iD]**, Yaohui Liu** [1,2][iD] **and Jingge Niu** [1]

[1] School of Surveying and Geo-Informatics, Shandong Jianzhu University, Jinan 250101, China; ymy@sdjzu.edu.cn (M.Y.); 2020165101@stu.sdjzu.edu.cn (X.C.); liuyaohui20@sdjzu.edu.cn (Y.L.); 2020160101@stu.sdjzu.edu.cn (J.N.)

[2] Hebei Key Laboratory of Earthquake Dynamics, Sanhe 065201, China

***** Correspondence: 2020160106@stu.sdjzu.edu.cn

**Abstract:** Automatic building extraction based on high-resolution aerial imagery is an important challenge with a wide range of practical applications. One of the mainstream methods for extracting buildings from high-resolution images is deep learning because of its excellent deep feature extraction capability. However, existing models suffer from the problems of hollow interiors of some buildings and blurred boundaries. Furthermore, the increase in remote sensing image resolution has also led to rough segmentation results. To address these issues, we propose a generative adversarial segmentation network (ASGASN) for pixel-level extraction of buildings. The segmentation network of this framework adopts an asymmetric encoder–decoder structure. It captures and aggregates multiscale contextual information using the ASPP module and improves the classification and localization accuracy of the network using the global convolutional block. The discriminator network is an adversarial network that correctly discriminates the output of the generator and ground truth maps and computes multiscale $L_1$ loss by fusing multiscale feature mappings. The segmentation network and the discriminator network are trained alternately on the WHU building dataset and the China typical cities building dataset. Experimental results show that the proposed ASGASN can accurately identify different types of buildings and achieve pixel-level high accuracy extraction of buildings. Additionally, compared to available deep learning models, ASGASN also achieved the highest accuracy performance (89.4% and 83.6% IoU on these two datasets, respectively).

**Keywords:** high-resolution aerial images; generative adversarial network; deep learning; WHU building dataset; China typical cities building dataset; semantic segmentation

## 1. Introduction

Automatic building extraction from high-resolution aerial imagery is of great significance in a wide range of application domains, such as disaster warning and processing, economic development assessment, and urban land use analysis [1–5]. In recent years, the improvement of remote sensing image resolution has made the spectral features of buildings more obvious and has provided richer semantic and texture features and other information for building extraction. However, high resolution can also lead to an increase in interference and redundant information. Therefore, the automatic extraction of buildings with high accuracy is a challenge to be tackled.

In the last few decades, to make full use of high-resolution remotely sensed imagery features for building information extraction, methods such as edge extraction, image segmentation, and digital morphology have been applied. These methods have achieved definite research results [6–10]. For example, the voting method was used to determine

the orientation of the house edges, and then least squares were used to refine the house edges [11]. Buildings can be extracted by a multi-scale segmentation method based on area growth, and the parameters of the best segmentation results are determined by multiple trials [12]. Building extraction can also be achieved by adding a priori knowledge of the building shape to the active contour model [13]. In addition, excellent machine learning classifiers have been used for the extraction of buildings, including boosting [14], support vector machines (SVM) [15] and random forest [16]. However, the above methods rely heavily on parameter selection and manually predefined features, leading to some limitations in practical building extraction.

With the rapid development of computer computing power and an increase in available data sources, deep learning techniques have been more widely and deeply developed, especially convolutional neural networks (CNNs) have played an important role in many fields [17]. Many scholars have used CNN models such as LeNet [18], AlexNet [19], VG-GNet [20], GoogleNet [21], and ResNet [22], which outperform traditional machine learning methods, to conduct related studies. However, CNNs perform region division and full convolution operations for each image element, which is computationally expensive. Meanwhile, the CNN processes the feature map into a fixed length output vector and outputs the result in the form of a numerical description [23]. Therefore, CNNs are not suitable for semantic segmentation tasks such as building extraction.

In 2015, Long proposed the Full Convolutional Network (FCN) for classification at the semantic level [24], which can be adapted to predict each pixel of the original image to accomplish the task of image semantic segmentation, and became the paradigm for many classical segmentation networks, mainly including the following. The first is (1) encoder–decoder neural networks. In the encoder part, the feature maps with smaller sizes are obtained by multiple downsampling to obtain multi-level semantic information, and in the decoder part, the final semantic segmentation maps are obtained by decoding the feature maps of each size. For example, The U-Net [25] method improves image segmentation accuracy by fusing multiscale information of images based on the symmetric coding structure through skip-connections. The SegNet [26] method is designed with a convolutional encoder with pooling and a decoder with deconvolution, which improves the edge portrayal and reduces the training parameters. The second is (2) image pyramid neural network, such as the DeepLab series models [27–30]. To better learn the features at different scales, DeepLabv3+ designs atrous spatial pyramid pooling (ASPP) to obtain more levels of semantic details, which effectively improves the segmentation accuracy. Meanwhile, these FCN-based models and their variants are widely used in tasks such as target extraction and image classification for remote sensing image processing. One scholar [31] added an attention mechanism with a multiple loss method to U-Net and successfully extracted buildings from publicly available aerial image datasets. Some scholars [32] improved the accuracy of small building detection based on residual features and feature pyramidal multiscale prediction. A team [33] proposed the RFA-U-Net model with a joint attention mechanism for building extraction. The FCN-based pixel-by-pixel prediction can ensure the accuracy of individual pixels, but the spatial relationship of the position before the pixel is often ignored.

In recent years, the emergence of generative adversarial networks (GANs) has provided an emerging solution thinking for semantic segmentation of building contour extraction. In our problem setup, the generator of the GAN is used to obtain a pixel-level semantic segmentation result map, while the discriminator is used to distinguish the segmentation map from the true coverage. The generator and discriminator are jointly optimized in the "adversarial" setting, so that the generator gets the best possible result. Methods based on GAN [34] can enhance the continuity of spatial labels and refine the segmentation results of the image. A team [35] used the DeepLabv3+ architecture as the basic segmentation model and the Pix2pix architecture as the GAN model to build a network that could perform the semantic segmentation task. Abdollahi et al. [36] propose a GAN model with SegNet and BCovLSTM modules for feature extraction and detection in complex environments,

which can effectively detect buildings obscured by features. To test the effectiveness of GAN application, Aunget et al. conducted building extraction experiments using Yangon city in Myanmar as a test area and obtained the local building footprint [37]. Although the above techniques made some achievements in solving the building extraction problem, the extracted building boundaries are not precise enough, and the accuracy can be further improved.

In order to further improve the accuracy and stability of the results of high-resolution building extraction and optimize the building boundaries more accurately, this paper proposes a new automatic building contour extraction network ASGASN. ASGASN has three main contributions:

(1) ASGASN implements automatic and efficient building segmentation based on GAN. In this algorithm, the segmentation network provides the class-false a priori knowledge for the training of the discriminator network, and the discriminator network corrects the learning of the segmentation network through training to make the classification results more closely match the a posteriori knowledge.

(2) The ASGASN architecture, into which ASPP and skip connections are embedded, allows features to be extracted from multiple spatial scales and improves segmentation accuracy by fusing multiscale information. The global convolutional block is also added to make a tight connection between the feature map and the pixel-by-pixel classifier.

(3) The segmentation and discriminator network are trained alternately by multiscale L1 loss and multiple cross entropy losses, which finally make the best performance of ASGASN. We conduct relevant experiments on both the WHU building dataset [38] and the Chinese typical city building dataset [39] to verify the advancedness of the present network.

## 2. Methods

### 2.1. Proposed Network ASGASN

The overall architecture of ASGASN is shown in Figure 1. This network follows the basic structure of GAN and is divided into segmentation network and discriminator network. The segmentation network is an asymmetric fully convolutional network, and the skip connection and ASPP are chosen as the bridge between the encoder and decoder to classify the input image at the semantic level. The discriminator network has a similar structure to the classical CNN network and aims at identifying the type of input, which has three inputs: original images, predicted results, and ground truth labels. The discriminator network learns potential higher-order feature structures by adversarial training and feeds them into the segmentation network to guide its learning. As learning proceeds, the accuracy of the discriminator network in identifying the truth of the input data increases. When the discriminator network has a strong performance but cannot identify whether the input is a prediction or ground truth, the segmentation network has a strong segmentation performance.

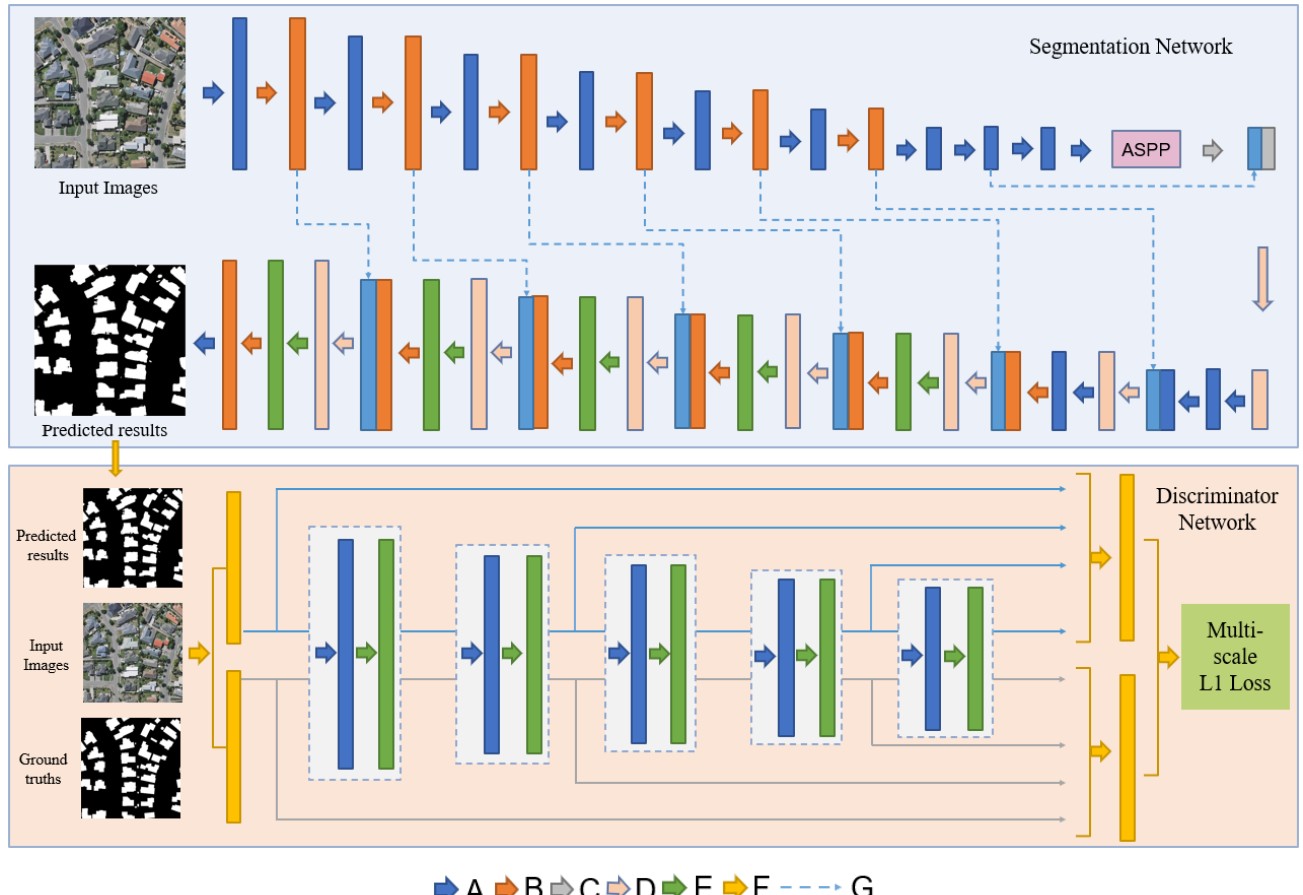

**Figure 1.** The structure of our proposed ASGASN. A are convolutional + batch normalization + Leaky rectified linear unit (Relu) layers; B is Residual block; ASPP is Atrous Spatial Pyramid Pooling; C is transposed convolution + batch normalization + Relu layers; D is upsample; E is Global convolutional block+ batch normalization+ Relu layers; F is concatenation operation; G is skip-connection.

## 2.2. Segmentation Network

The segmentation network of ASGASN is a horizontal U-shaped asymmetric deep learning model, with the upper end performing image input and the lower end directly outputting building extraction results. The detailed structure is given in Figure 2 and Table 1. As we can see from Table 1, it contains eight encoder blocks, one ASPP module and nine decoder blocks. The encoder (blocks 1–8) consists of convolutional blocks and several residual blocks that can extract feature mappings at different levels from global and local contextual information. The decoder (blocks 10–18) consists of convolutional blocks, residual blocks and global convolutional blocks. Convolutional blocks are used to resize feature maps and the change the number of channels, residual blocks are used to deepen the network layers to prevent network degradation, and upsampling is used to recover the image details. A skip connection fuses each upsampled mapping with the corresponding size feature mapping in the encoder. The fusion output is sequentially and progressively fused with the multilevel feature map and restored to the size of the original input map. Finally, there is an output layer after the encoder that performs pixelwise classification. ASPP (block 9) is used as a bridge between upsampling and downsampling, while extracting multiple scales of semantic information from the small size feature map, so that better global learning of the feature map can be performed. ASGASN's detailed blocks are shown in Table 1: 'Conv' stands for Convolutional, 'ReLU' stands for Rectified Linear Unit, 'BN' stands for Batch Normalization, and 'GlobalConv' denotes Global Convolutional.

**Table 1.** The detailed blocks of the proposed ASGASN.

| Block | Type | Kernel Size | Input | Output |
|:-----:|:----:|:-----------:|:-----:|:------:|
| | Conv1 | (7, 7) | $3 \times 128 \times 128$ | $64 \times 64 \times 64$ |
| 1 | LeakyReLU1 | | $64 \times 64 \times 64$ | $64 \times 64 \times 64$ |
| | Residule Block1 | | $64 \times 64 \times 64$ | $64 \times 64 \times 64$ |
| | Conv2 | (5, 5) | $64 \times 64 \times 64$ | $128 \times 32 \times 32$ |
| 2 | BN1 + LeakyReLU2 | | $128 \times 32 \times 32$ | $128 \times 32 \times 32$ |
| | Residule Block2 | | $128 \times 32 \times 32$ | $128 \times 32 \times 32$ |
| | Conv3 | (5, 5) | $128 \times 32 \times 32$ | $256 \times 16 \times 16$ |
| 3 | BN2 + LeakyReLU3 | | $256 \times 16 \times 16$ | $256 \times 16 \times 16$ |
| | Residule Block3 | | $256 \times 16 \times 16$ | $256 \times 16 \times 16$ |
| | Conv4 | (5, 5) | $256 \times 16 \times 16$ | $512 \times 8 \times 8$ |
| 4 | BN3 + LeakyReLU4 | | $512 \times 8 \times 8$ | $512 \times 8 \times 8$ |
| | Residule Block4 | | $512 \times 8 \times 8$ | $512 \times 8 \times 8$ |
| | Conv5 | (5, 5) | $512 \times 8 \times 8$ | $512 \times 4 \times 4$ |
| 5 | BN4 + LeakyReLU5 | | $512 \times 4 \times 4$ | $512 \times 4 \times 4$ |
| | Residule Block5 | | $512 \times 4 \times 4$ | $512 \times 4 \times 4$ |
| | Conv6 | (4, 4) | $512 \times 4 \times 4$ | $1024 \times 2 \times 2$ |
| 6 | BN5 + LeakyReLU6 | | $1024 \times 2 \times 2$ | $1024 \times 2 \times 2$ |
| | Conv7 | (1, 1) | $1024 \times 2 \times 2$ | $1024 \times 2 \times 2$ |
| 7 | Conv8 | (3, 3) | $1024 \times 2 \times 2$ | $2048 \times 1 \times 1$ |
| | BN6 + LeakyReLU7 | | $2048 \times 1 \times 1$ | $2048 \times 1 \times 1$ |
| 8 | Conv9 | (1, 1) | $2048 \times 1 \times 1$ | $512 \times 1 \times 1$ |
| | BN7 + LeakyReLU8 | | $512 \times 1 \times 1$ | $512 \times 1 \times 1$ |
| 9 | ASPP | | $512 \times 1 \times 1$ | $512 \times 1 \times 1$ |
| | Conv10 | (1, 1) | $512 \times 1 \times 1$ | $2048 \times 1 \times 1$ |
| 10 | BN8 + ReLU1 | | $2048 \times 1 \times 1$ | $2048 \times 1 \times 1$ |
| | Upsample1 | | $2048 \times 1 \times 1$ | $1024 \times 2 \times 2$ |
| | Conv11 | (3, 3) | $1024 \times 2 \times 2$ | $1024 \times 2 \times 2$ |
| | BN9 + ReLU2 | | $1024 \times 2 \times 2$ | $1024 \times 2 \times 2$ |
| 11 | Conv12 | (1, 1) | $1024 \times 2 \times 2$ | $2048 \times 2 \times 2$ |
| | BN10 + ReLU3 | | $2048 \times 2 \times 2$ | $2048 \times 2 \times 2$ |
| | Upsample2 | | $2048 \times 2 \times 2$ | $2048 \times 4 \times 4$ |
| | Conv12 | (3, 3) | $2048 \times 4 \times 4$ | $512 \times 4 \times 4$ |
| 12 | BN10 + ReLU3 | | $512 \times 4 \times 4$ | $512 \times 4 \times 4$ |
| | Residule Block6 | | $512 \times 4 \times 4$ | $512 \times 4 \times 4$ |
| | Upsample3 | | $512 \times 4 \times 4$ | $1024 \times 8 \times 8$ |
| | GlobalConv Block1 | | $1024 \times 8 \times 8$ | $512 \times 8 \times 8$ |
| 13 | BN11 + ReLU4 | | $512 \times 8 \times 8$ | $512 \times 8 \times 8$ |
| | Residule Block7 | | $512 \times 8 \times 8$ | $512 \times 8 \times 8$ |
| | Upsample4 | | $512 \times 8 \times 8$ | $1024 \times 16 \times 16$ |
| | GlobalConv Block2 | | $1024 \times 16 \times 16$ | $256 \times 16 \times 16$ |
| 14 | BN12 + ReLU5 | | $256 \times 16 \times 16$ | $256 \times 16 \times 16$ |
| | Residule Block8 | | $256 \times 16 \times 16$ | $256 \times 16 \times 16$ |
| | Upsample5 | | $256 \times 16 \times 16$ | $512 \times 32 \times 32$ |
| | GlobalConv Block3 | | $512 \times 32 \times 32$ | $128 \times 32 \times 32$ |
| 15 | BN13 + ReLU6 | | $128 \times 32 \times 32$ | $128 \times 32 \times 32$ |
| | Residule Block9 | | $128 \times 32 \times 32$ | $128 \times 32 \times 32$ |
| | Upsample6 | | $128 \times 32 \times 32$ | $256 \times 64 \times 64$ |
| | GlobalConv Block4 | | $256 \times 64 \times 64$ | $64 \times 64 \times 64$ |
| 16 | BN14 + ReLU7 | | $64 \times 64 \times 64$ | $64 \times 64 \times 64$ |
| | Residule Block10 | | $64 \times 64 \times 64$ | $64 \times 64 \times 64$ |
| | Upsample7 | | $64 \times 64 \times 64$ | $128 \times 128 \times 128$ |
| | GlobalConv Block5 | | $128 \times 128 \times 128$ | $64 \times 128 \times 128$ |
| 17 | BN15 + ReLU8 | | $64 \times 128 \times 128$ | $64 \times 128 \times 128$ |
| | Residule Block11 | | $64 \times 128 \times 128$ | $64 \times 128 \times 128$ |
| | Upsample8 | | $64 \times 128 \times 128$ | $64 \times 128 \times 128$ |
| 18 | Conv13 | (5, 5) | $64 \times 128 \times 128$ | $3 \times 128 \times 128$ |

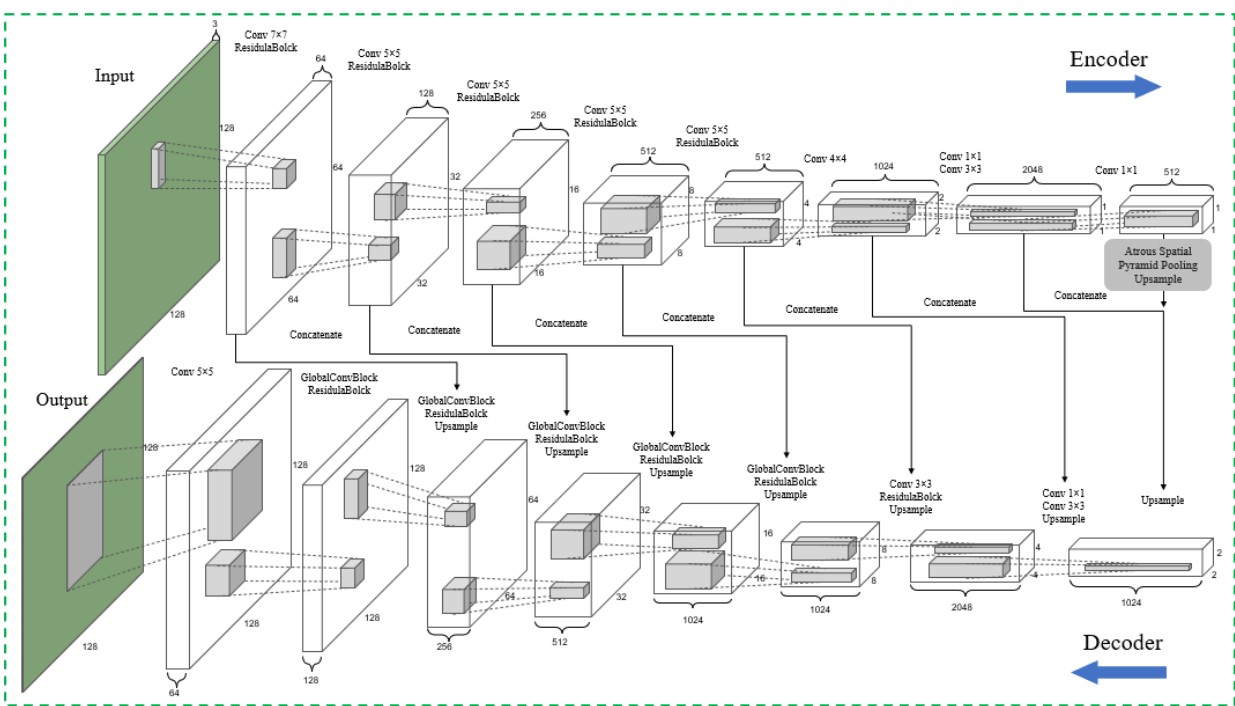

**Figure 2.** Structure of Segmentation Network.

## 2.2.1. Subsubsection Atrous Spatial Pyramid Pooling

The ASPP draws from DeepLabv3+ [29]. The details of the ASPP structure are shown in Figure 3. It has several parallel atrous convolutions containing different rete, and the final output is performed by feature map fusion. Four different rates of atrous convolutions operate in parallel to ensure better retention of semantic information in images while keeping the network computation constant. After the ASPP model, different scale feature maps containing rich semantic features can be obtained, and feature map fusion is finally performed to effectively improve the feature sensitivity of the model [40]. In this network, we employed the ASPP module with image pooling and $1 \times 1$ convolution and three-branches of atrous convolution with different rates to effectively capture multiscale contextual information.

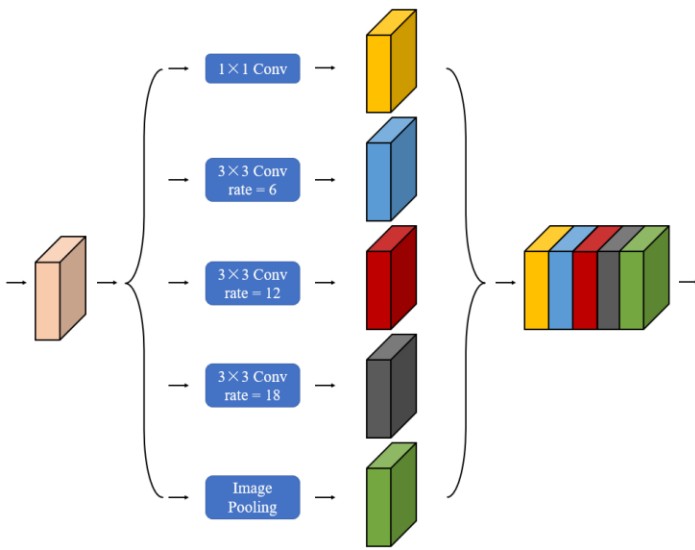

**Figure 3.** The structure of atrous spatial pyramid pooling.

### 2.2.2. Residual Block

The residual block was proposed by He et al. [41], and a good solution to the phenomenon that increasing the network depth tends to cause gradient dispersion and gradient explosion. Figure 4 shows the specific structure of the residual block. The input image feature maps are first passed through convolutional modules with $1 \times 1$, $3 \times 3$, and $1 \times 1$ convolutional kernels and then fused with the initial feature maps by skip connection to obtain new feature maps for output. In this work, we designed the residual block into the segmentation network to help improve network performance and prevent network degradation.

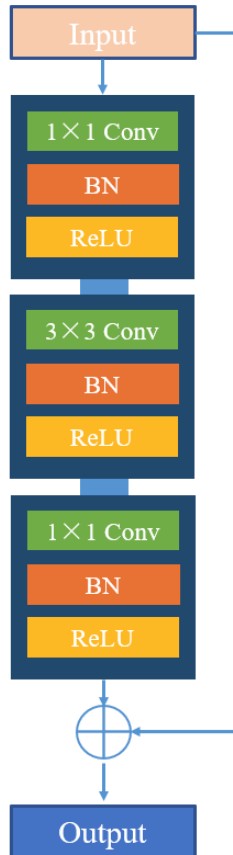

**Figure 4.** The structure of residual block.

### 2.2.3. Global Convolutional Block

Semantic segmentation for extracting buildings based on remote sensing images requires intensive pixel prediction when both classification and localization tasks need to be performed. As shown in Figure 5b, the traditional semantic segmentation approach focuses on the localization problem, but this may reduce the classification performance. We follow the idea of global convolutional [42] and consider two points: from the localization point of view, the model should use full convolution to maintain the localization performance; from the classification point of view, a larger-size kernel should be used in the network structure to make a tight connection between the feature map and the pixel-by-pixel classifier. Therefore, the global convolutional block is used in the segmentation network to improve the classification and localization accuracy of the network. The details of the global convolutional block structure are shown in Figure 5d; it uses a two-branch structure, with the left branch using $k \times 1 + 1 \times k$ convolution processing, while the right branch uses a combination of $1 \times k + k \times 1$ for the final feature map fusion to ensure sufficient

processing in the k × k region. It also ensures a reduction in computation with a certain feeling of wildness.

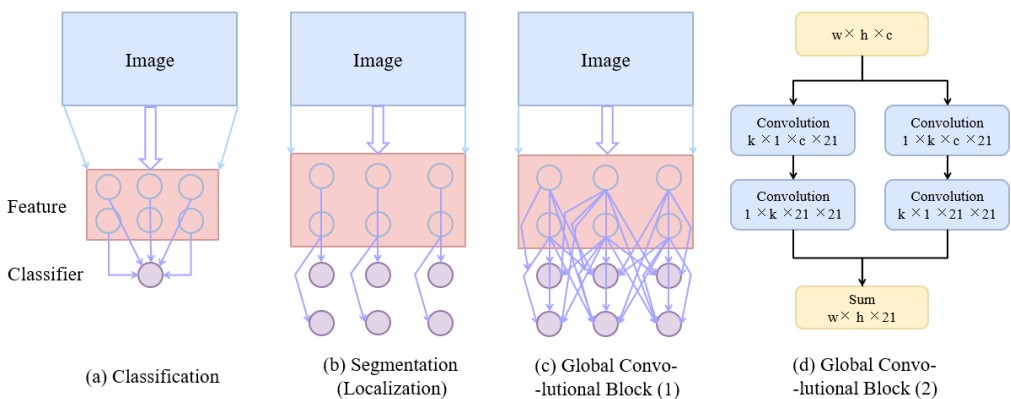

(a) Classification     (b) Segmentation (Localization)     (c) Global Convo-lutional Block (1)     (d) Global Convo-lutional Block (2)

**Figure 5.** (**a**): Classification network; (**b**): Conventional segmentation network, mainly designed for localization; (**c**): Global Convolutional Block; (**d**): The structure of Global Convolutional Block.

### 2.3. Discriminator Network

There are two kinds of inputs in the discriminator network: original images fused with the predicted results mapping and original images fused with the ground truths. Figure 6 illustrates the specific structure of the discriminatory network. It first goes through five convolutional and global convolutional blocks, then follows convolution, batch normalization, and ReLU layers, and finally generates a distribution map through a convolution-sigmoid layer. To better obtain the spatial and positional relationships between pixels, we obtain global information by fusing semantic feature maps at different levels. Finally, we adjust the learning and training of the network by calculating multiscale $L_1$ loss for back propagation.

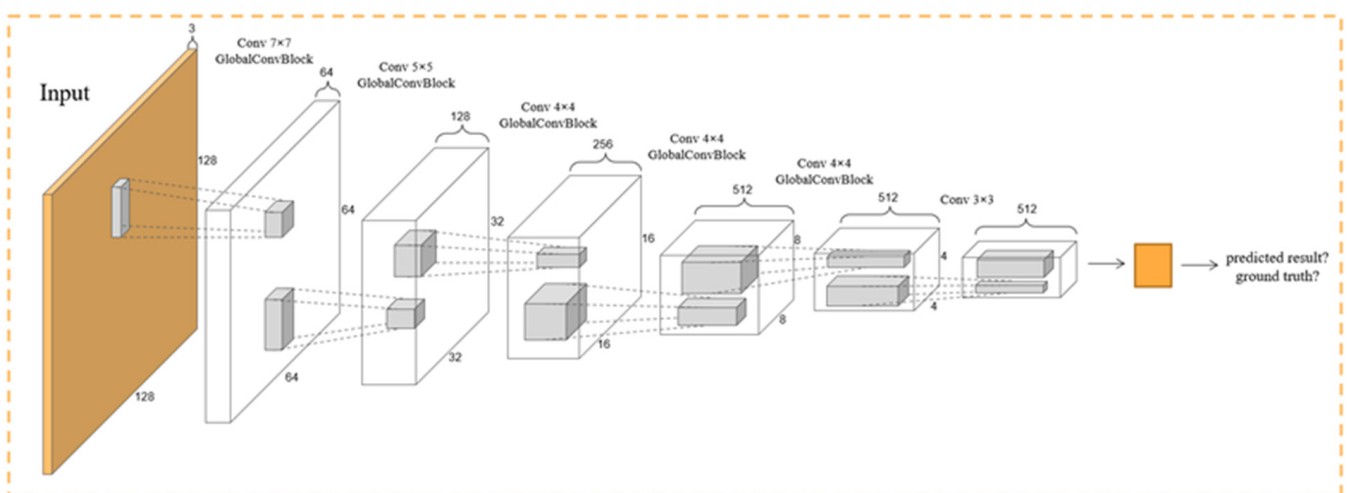

**Figure 6.** Structure of discriminator Network.

### 2.4. Loss Function

In order to optimize the performance of ASGASN, we train the network by a dual loss function. The discriminator network is first trained with multi-scale $L_1$ loss, and the segmentation network is trained using cross-entropy loss on the basis of back-propagation. The multi-scale $L_1$ loss is defined as:

$$L_C = -\frac{1}{N} \sum_{n=1}^{N} l_{mae}(f_C(x_n, S(x_n)), f_C(x_n, y_n)) \tag{1}$$

where $(x_n, S(x_n))$ is the concatenation of original images and predicted result, $y_n$ is the ground truth, $f_C(x)$ denotes the semantic features obtained from $x$ at different levels, $l_{mae}$ is the $L_1$ distance. $l_{mae}$ is defined as:

$$l_{mae}(f_C(x), f_C(x')) = \frac{1}{L} \sum_{i=1}^{L} \| f_C^i(x) - f_C^i(x') \|_1 \tag{2}$$

where $L$ represents the sum of all feature sizes in the segmentation, $f_C^i(x)$ denotes the features characteristics in a single scale $i$.

$$L_S = -\frac{1}{N}(y(x_n)log(S(x_n)) + (1 - y(x_n))log(1 - S(x_n))) - L_C \tag{3}$$

where $y(x_n)$ denotes the true distribution of the ground in the nth remote sensing image.

### 2.5. Flowchart

The training and testing process of building segmentation based on ASGASN is shown in Figure 7. The left side of the dashed line shows the training process of ASGASN. First, the original image and the ground truth are input to the segmentation network to obtain the corresponding prediction results. Then, the discriminator network discriminates the ground truth from the predicted results. In the discriminator network, the multiscale $L_1$ loss is obtained by computing the difference between the predicted result and the ground truth. Meanwhile, multiscale $L_1$ loss and cross-entropy losses are back-propagated using gradient descent, and the parameters of the whole network are continuously updated until the training is completed. The right side of the dashed line shows the testing process of ASGASN. In the testing process, only the original images need to be input into the trained segmentation network to obtain the building extraction results.

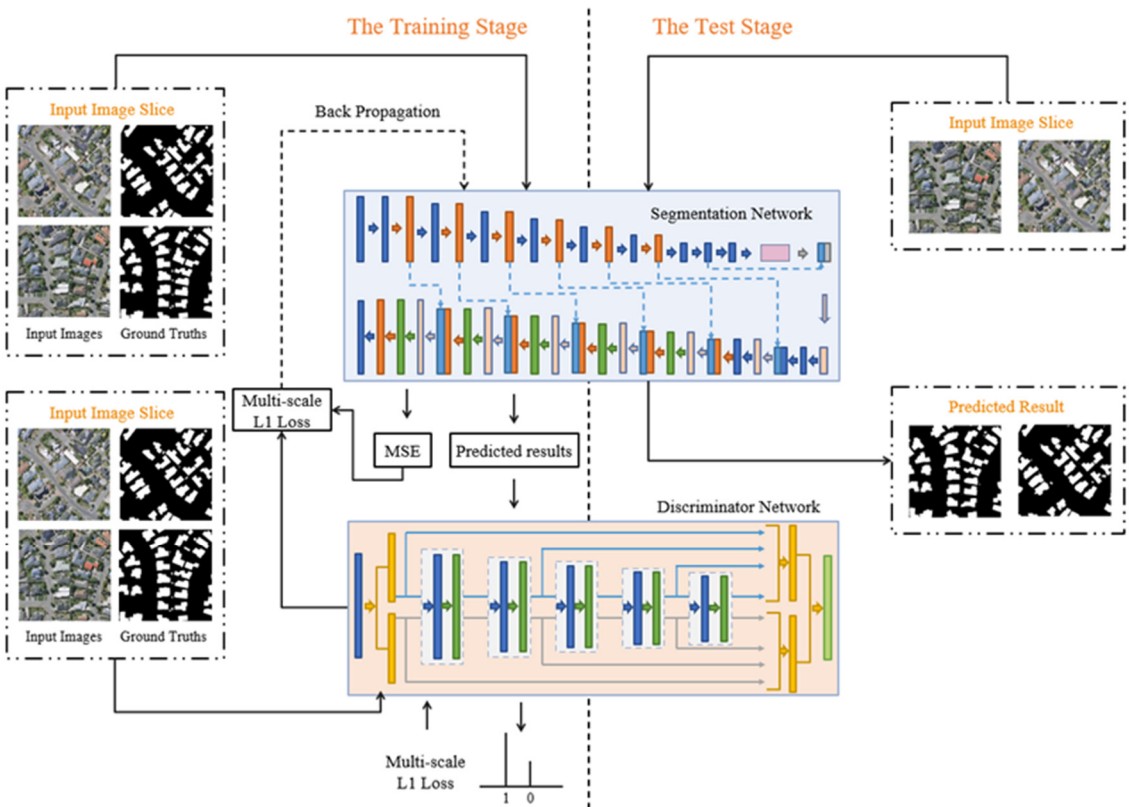

**Figure 7.** Flowchart of building extraction from high-resolution remote sensing image based on ASGASN.

### 2.6. Pixel Analysis

The ASGASN proposed in this paper is a pixel-level building extraction of the high-resolution aerial imagery, that is, a category attribution for each pixel on the image. The input of the network is a cut subgraph of the training data after data enhancement, and the output image has the same size as the input image, and the value of each pixel point indicates the predicted category value of that point. This pixel-level analysis allows for more accurate extraction of building contour information, such as small depressions or protrusions. It also improves the recognition of small-sized buildings, so that the different types of buildings in the whole image can be acquired more completely. In addition, some open-source software such as QGIS is also based on pixel analysis for snow mapping and dew volume estimation [43,44].

### 3. Experiment Dataset and Evaluation

### 3.1. Experiment Data

The first dataset used in this study is the WHU building dataset [38] from the New Zealand Land Information Service website. The WHU dataset covers 450 km$^2$ of land on the ground and was selected from approximately 22,000 individual buildings in the Christchurch area with a spatial resolution of 0.3 m. The dataset provides shapefile format data of buildings as well as rasterized data. Figure 8 is an example of the original images and their corresponding ground truths. White represents buildings, and black represents the background. The second dataset used in this study is the China typical cities (CHN) building dataset [39]. The original data were derived from Google's Class 19 satellite imagery with a ground resolution of 0.29 m. The sample dataset covers a total area of approximately 120 km$^2$. This dataset contains a sample of 7260 image areas, with a total of 63,886 buildings, distributed among four cities: Beijing, Shanghai, Shenzhen and Wuhan. Figure 9 shows the original image from the dataset and its corresponding ground truth.

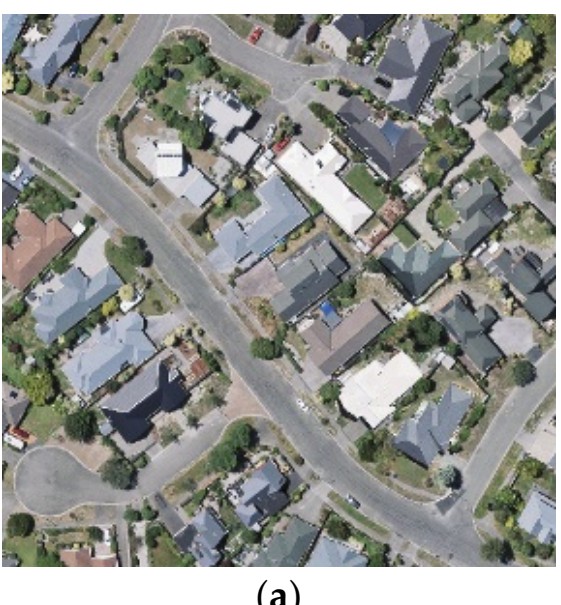
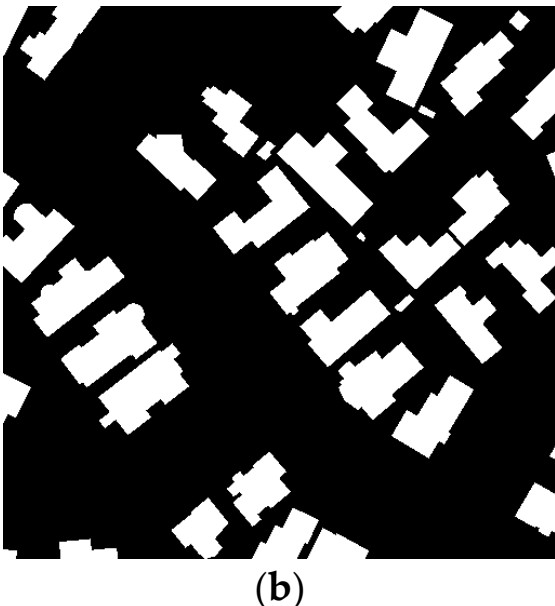

(**a**)　　　　　　　　　　　　　　　　(**b**)

**Figure 8.** Image and label example selected from the WHU dataset: (**a**) Original image; (**b**) Ground truth. Black and white pixels mark non-building and building, respectively.

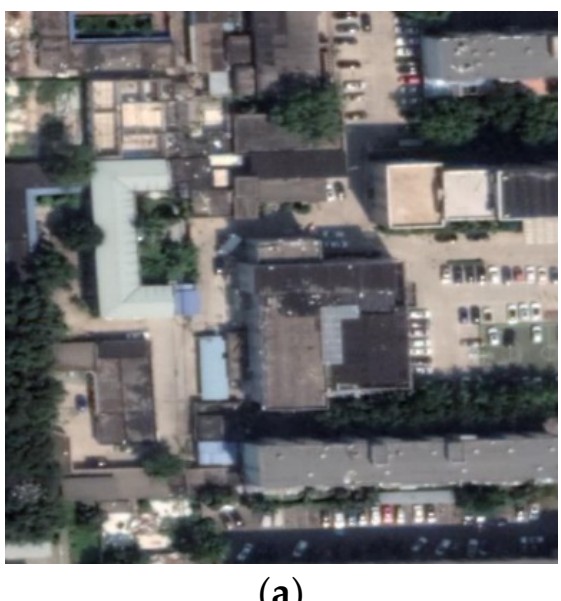

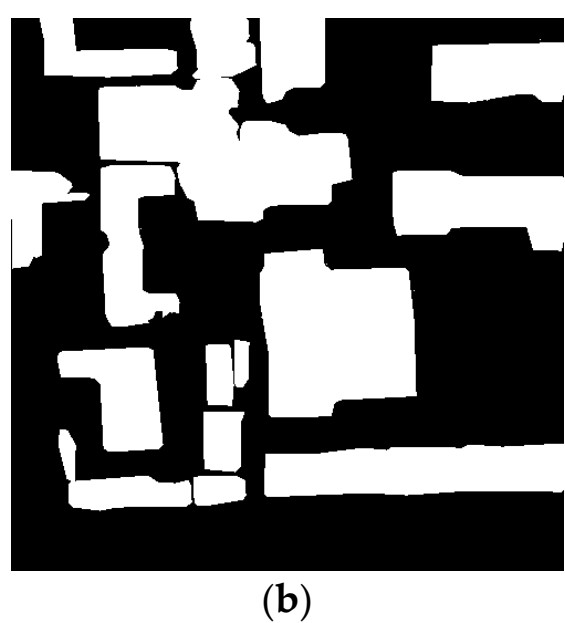

**(a)** **(b)**

**Figure 9.** Image and label example selected from the China typical cities building dataset: (**a**) Original image; (**b**) Ground truth. Black and white pixels mark non-building and building, respectively.

### 3.2. Data Processing

The data enhancement method increases the training sample by sample expansion to avoid overfitting of the model [45]. In this study, the samples were subjected to vertical and horizontal mirror flips as well as rotations of different angles. Figure 10 shows an example of the imagery after data processing and enhancement.

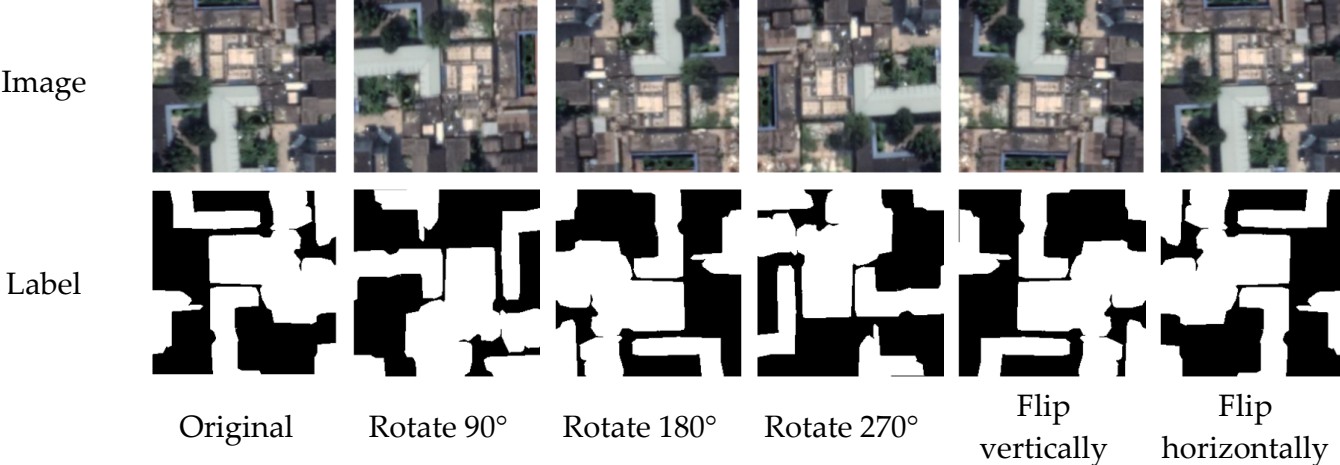

**Figure 10.** An example of data augmentation by rotating and flipping.

### 3.3. Experiment Settings

This experiment was conducted based on the PyTorch deep learning framework with an NVIDIA GeForce RTX 3070 graphics card. The images in the dataset were randomly cropped to 256 × 256 pixels to better utilize the power of the GPU and improve computational efficiency. In the parameter setting, the most suitable model parameters were finally determined through several controlled variable experiments. To avoid the imbalance caused by the strong discriminator network between the two models, the optimization goal of the discriminator network was divided by two to reduce the learning speed of the segmentation network. To accommodate the memory limitation of the computer GPU, the

model was trained by inputting eight images per batch and training 200 epochs to obtain the optimal model parameters for each of the two data sets.

### 3.4. Evaluation Metrics

In this paper, we experimentally test the validity and accuracy of each model and evaluate the model performance based on several metrics: the 'overall accuracy' (OA), 'precision', 'recall', 'F1-score', and intersection over union ('IoU'). The five metrics are presented as follows:

1. OA refers to the proportion of correctly predicted building and background pixels to all pixels in the image:

$$OA = \frac{TP + TN}{TP + TN + FP + FN} \tag{4}$$

where TP represents the number of buildings extracted as buildings and the actual number of buildings; FP represents the number of buildings extracted as buildings and the actual number of backgrounds; TN represents the number of buildings extracted as backgrounds and the actual number of backgrounds; and FN represents the number of buildings extracted as backgrounds and the actual number of buildings.

2. Recall refers to the proportion of correctly predicted building pixels in the image to the true value pixels in the building area:

$$Precision = \frac{TP}{TP + FP} \tag{5}$$

3. Precision refers to proportion of correctly predicted building pixels to all predicted building pixels in the image:

$$Recall = \frac{TP}{TP + FN} \tag{6}$$

4. F1-score represents the weighted average of OA and Precision:

$$F1 = \frac{2}{\left(\frac{1}{Precision}\right) + \left(\frac{1}{Recall}\right)} \tag{7}$$

5. IoU, which can describe segment-level accuracy:

$$IoU = \frac{TP}{TP + FP + FN} \tag{8}$$

### 3.5. Model Comparisons

FCN8s: FCN is based on VGG and completes the conversion from a classification model to a semantic segmentation model by replacing the fully concatenated layer with an inverse convolution layer [23], pioneering the application of fully convolutional networks to image segmentation. The FCN upsamples the feature maps obtained from the convolution layer and classifies them for each pixel while recovering the feature map size.

PSPNet: Zhao et al. [46] proposed a pyramid pooling module (PSPNet) that aggregates contextual information based on different regions and has the ability to mine global contextual information. Based on the pixel-by-pixel prediction, PSPNet also mines the global information of the remote sensing imagery to improve the accuracy of predicting features through multi-scale information aggregation.

SegNet: SegNet [25] is an FCN-based encoder–decoder structured semantic segmentation network with a max pooling operation. Such a structural design allows the network to upsample the underlying information input feature maps. Therefore, SegNet has better performance and efficiency in semantic segmentation.

U-Net: U-Net [24] is a classical architecture with a symmetric encoder–decoder structure. The compressed paths in U-Net are used to better capture feature information and features, and their symmetric extended paths can restore feature map size, while fusion of feature maps is performed by jumping connections to retain the maximum amount of important feature information. Due to its good robustness, U-Net has been widely used as a base framework for many segmentation models in recent years.

## 4. Results

### 4.1. Experimental Results on the WHU Dataset

In order to verify the effectiveness of the model for buildings, this paper conducted building extraction experiments based on the WHU dataset and FCN8s, PSPNet, SegNet, and U-Net models, and the extraction results are shown in Figure 11. Overall, PSPNet returns more false negatives (blue) and the fewest true positive (green), while FCN8s got the most false positives (red). Although both SegNet and U-Net return more positives (green) results, SegNet obtains more false negatives (blue) than U-Net. In contrast, ASGASN shows significantly fewer false positives (red) and false negatives (blue) than the other models and the extracted building outline is closer to the real situation on the ground.

The following is a comparative analysis of the extraction results for different types of buildings. The red boxes in the first column show the extraction results of different models for irregular buildings. FCN8s, PSPNet, and SegNet only extracted a part of the buildings, and a considerable part was not detected. The U-Net model extracts the integrity of buildings better than these three models but still suffers from the problem of fine blurred edges. ASGASN extraction is relatively complete, with clear building boundaries and basically no voids. The red box in the second column shows the extraction of buildings at the edge of the image by different models. FCN8s, PSPNet, SegNet, and U-Net poorly extract buildings, all failing to accurately extract the boundaries of buildings and, in some cases, not detecting buildings at all. ASGASN can detect all buildings; although there are a few false positives (red), ASGASN still obtains a smoother edge for the building outline. The red box in the third column shows the extraction of different models for small-scale buildings. FCN8s, PSPNet, SegNet, and U-Net are basically unable to extract buildings and only extract sporadic patches in the interior. ASGASN can detect buildings at smaller scales and extract a higher degree of completeness. The red box in the fourth column shows the extraction of regular buildings by different models. The PSPNet building extraction is the worst as basically no edge details are detected. FCN8s, SegNet, and U-Net extracted buildings have more accurate edges, but still result in false negatives (blue). ASGASN extracts the edges of the building outline more smoothly and accurately, and the integrity of the building interior is higher. Therefore, ASGASN achieves the best extraction results.

Table 2 shows the comparison of the results of all models for each evaluation metric, and ASGASN achieved the highest scores in four metrics. Among them, the OA of ASGASN is 2.3% better than SegNet, which has the highest accuracy among the other models. For recall, the performance of U-Net and the proposed ASGASN is significantly better than that of the other three methods. ASGASN has the highest performance, which is 4.4% higher than that of U-Net. The ASGASN F1-score reached 94.4%, which is 3.4%, 5.3%, 13.0% and 5.3% higher than those of U-Net, SegNet, PSPNet and FCN8s, respectively. The cross-convergence ratios increased by 3.5%, 9.2%, 17.2% and 9.1% relative to U-Net, SegNet, PSPNet and FCN8s, respectively.

**Table 2.** Quantitative comparison with the state-of-the-art models on the WHU dataset. The highest value for each metric is marked as bold.

| Metrics | Methods | Image1 | Image2 | Image3 | Image4 | Mean |
|---------|---------|--------|--------|--------|--------|------|
| OA | FCN8s | 0.938 | 0.945 | 0.963 | 0.938 | 0.946 |
| | PSPNet | 0.931 | 0.899 | 0.927 | 0.906 | 0.915 |
| | SegNet | 0.951 | 0.938 | 0.973 | 0.944 | 0.951 |
| | U-Net | 0.973 | 0.953 | 0.969 | 0.959 | 0.936 |
| | ASGASN | **0.977** | **0.971** | **0.981** | **0.968** | **0.974** |
| Precision | FCN8s | **0.967** | 0.953 | 0.876 | 0.925 | 0.931 |
| | PSPNet | 0.944 | 0.921 | 0.924 | 0.926 | 0.928 |
| | SegNet | 0.954 | 0.955 | **0.954** | 0.957 | **0.955** |
| | U-Net | 0.938 | **0.961** | 0.913 | **0.959** | 0.942 |
| | ASGASN | 0.947 | 0.937 | 0.936 | 0.932 | 0.938 |
| Recall | FCN8s | 0.785 | 0.874 | 0.897 | 0.871 | 0.856 |
| | PSPNet | 0.779 | 0.787 | 0.703 | 0.787 | 0.764 |
| | SegNet | 0.804 | 0.828 | 0.864 | 0.836 | 0.833 |
| | U-Net | 0.939 | 0.889 | 0.899 | 0.901 | 0.907 |
| | ASGASN | **0.948** | **0.962** | **0.936** | **0.955** | **0.951** |
| F1-score | FCN8s | 0.867 | 0.912 | 0.887 | 0.897 | 0.891 |
| | PSPNet | 0.854 | 0.849 | 0.703 | 0.851 | 0.814 |
| | SegNet | 0.873 | 0.887 | 0.907 | 0.893 | 0.891 |
| | U-Net | 0.939 | 0.923 | 0.906 | 0.929 | 0.924 |
| | ASGASN | **0.947** | **0.951** | **0.936** | **0.943** | **0.944** |
| IoU | FCN8s | 0.765 | 0.838 | 0.797 | 0.814 | 0.803 |
| | PSPNet | 0.745 | 0.737 | 0.665 | 0.741 | 0.722 |
| | SegNet | 0.775 | 0.797 | 0.831 | 0.806 | 0.802 |
| | U-Net | 0.885 | 0.858 | 0.828 | 0.867 | 0.859 |
| | ASGASN | **0.901** | **0.904** | **0.881** | **0.893** | **0.894** |

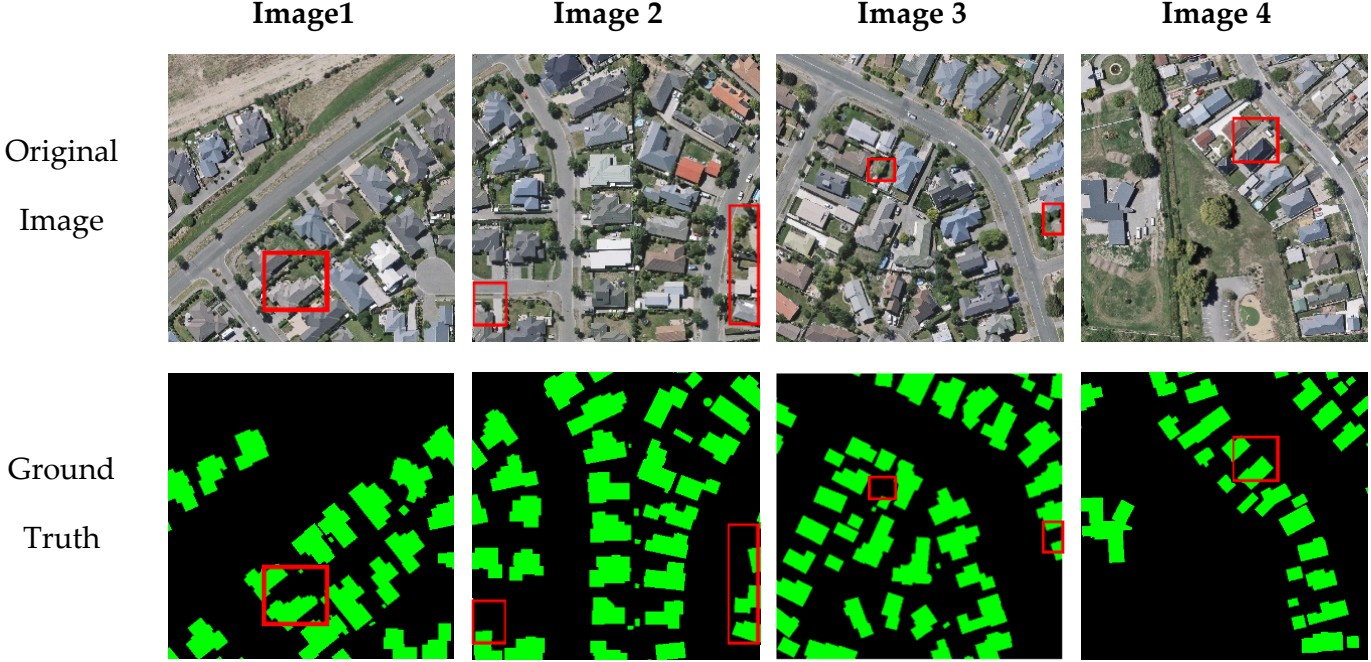

**Figure 11.** *Cont*.

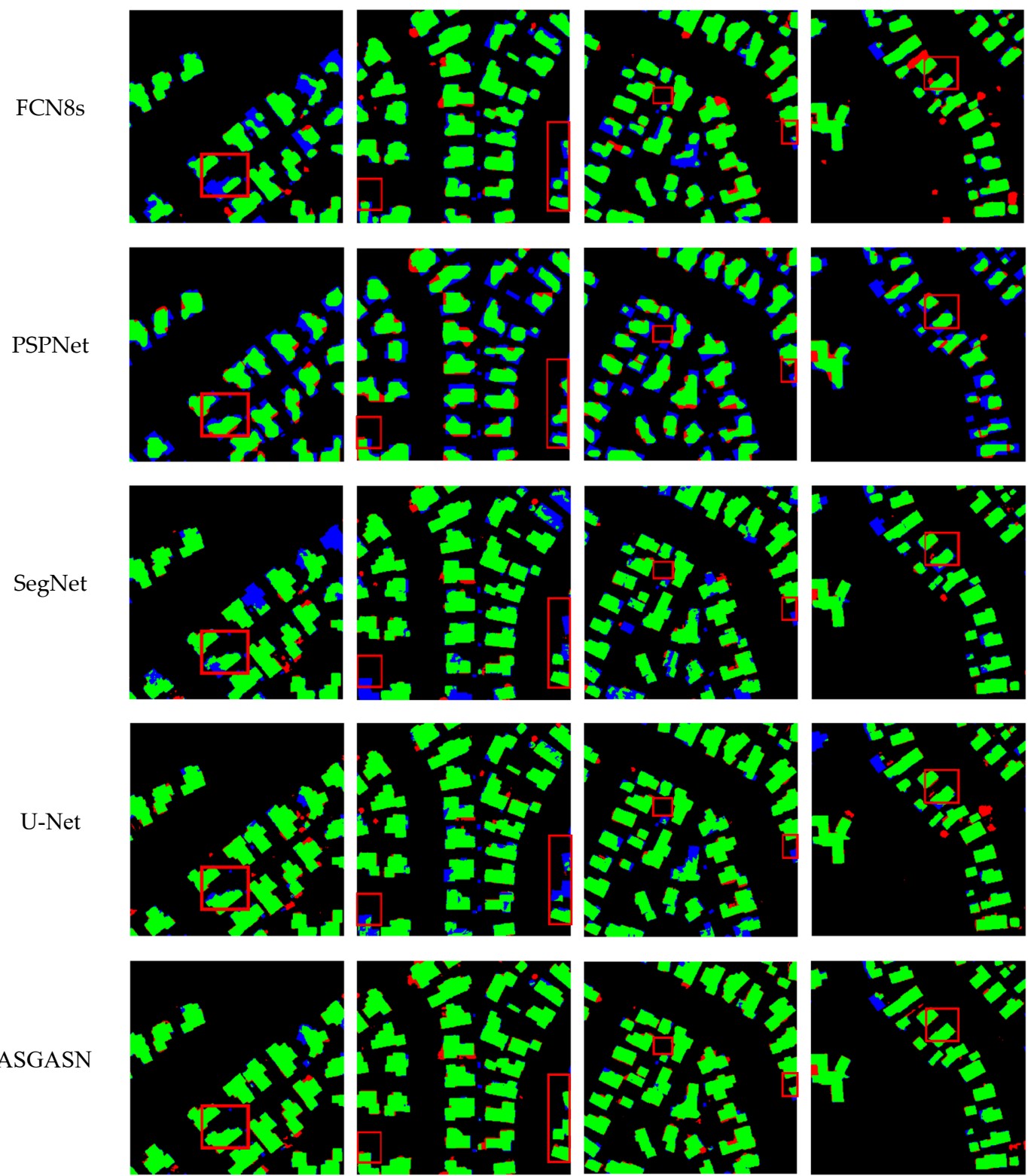

**Figure 11.** Examples of segmentation results by different models on the WHU dataset. The green, red, blue, and black pixels of the maps represent the true positive, false positive, false negative, and true negative predictions, respectively.

## 4.2. Experimental Results on the CHN Dataset

Figure 12 shows the building extraction results of different models on the CHN dataset. Overall, ASGASN, U-Net and SegNet have significantly higher performance than FCN8s and PSPNet. FCN8s and PSPNet can only obtain the general location of the building, and have difficulty in extracting detailed information of buildings. In particular, for Image 4, both models return too many false positives (blue), indicating that there are missing buildings in the extraction results. Between ASGASN, U-Net and SegNet, ASGASN returns the most negatives (green) and has basically no voids in the extracted buildings. In contrast, ASGASN gives a more accurate building profile and no voids in the results. In order to evaluate the performance of each network more objectively, we calculated their evaluation metrics in this test set as shown in Table 3. Compared to other models, ASGASN has the highest scores on all four metrics. Among them, U-Net has the highest performance apart from ASGASN. Compared to U-Net, ASGASN improved 0.6% in F1 score (0.905 vs. 0.911) and 0.9% in IoU (0.827 vs. 0.836).

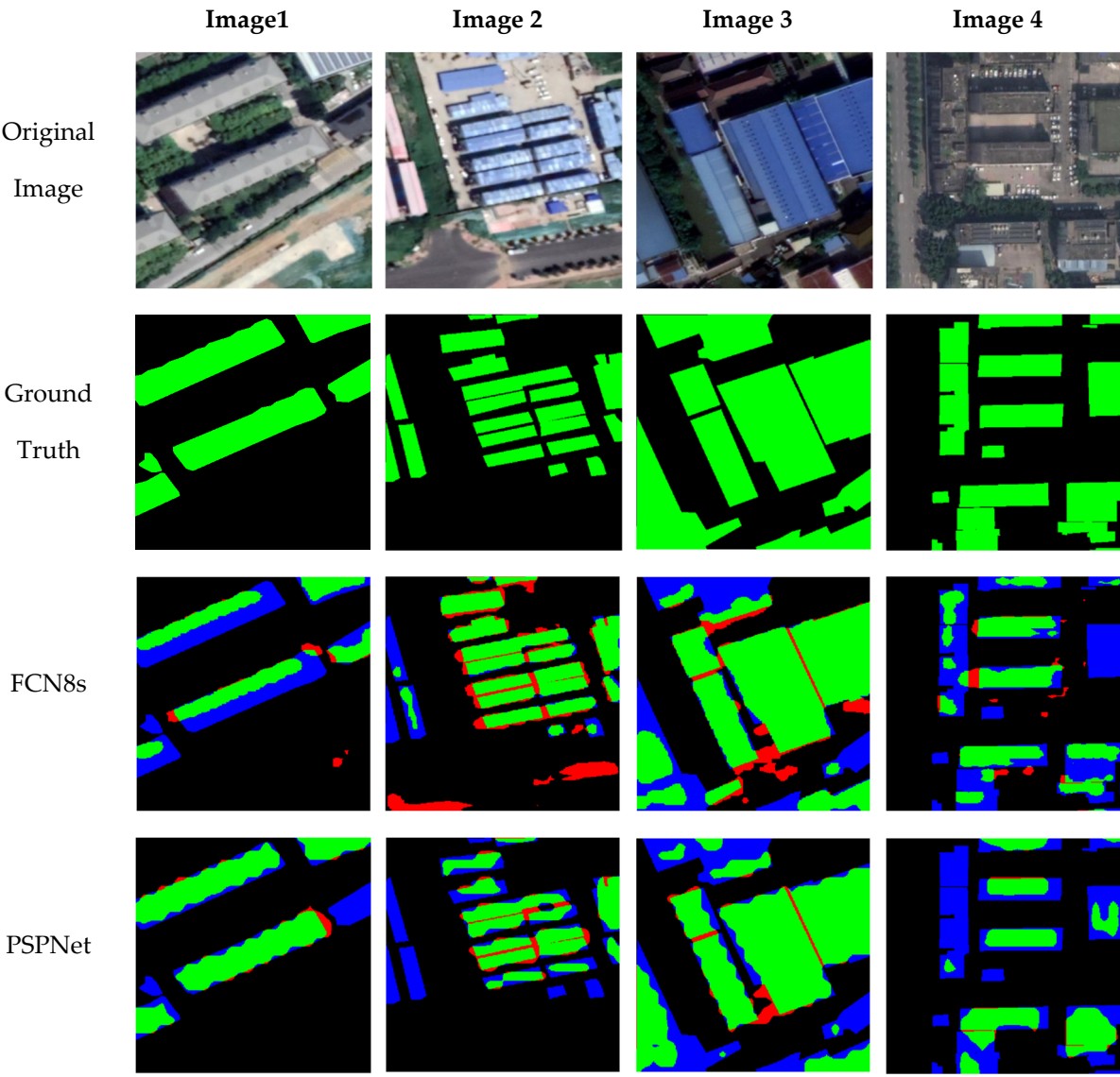

**Figure 12.** *Cont.*

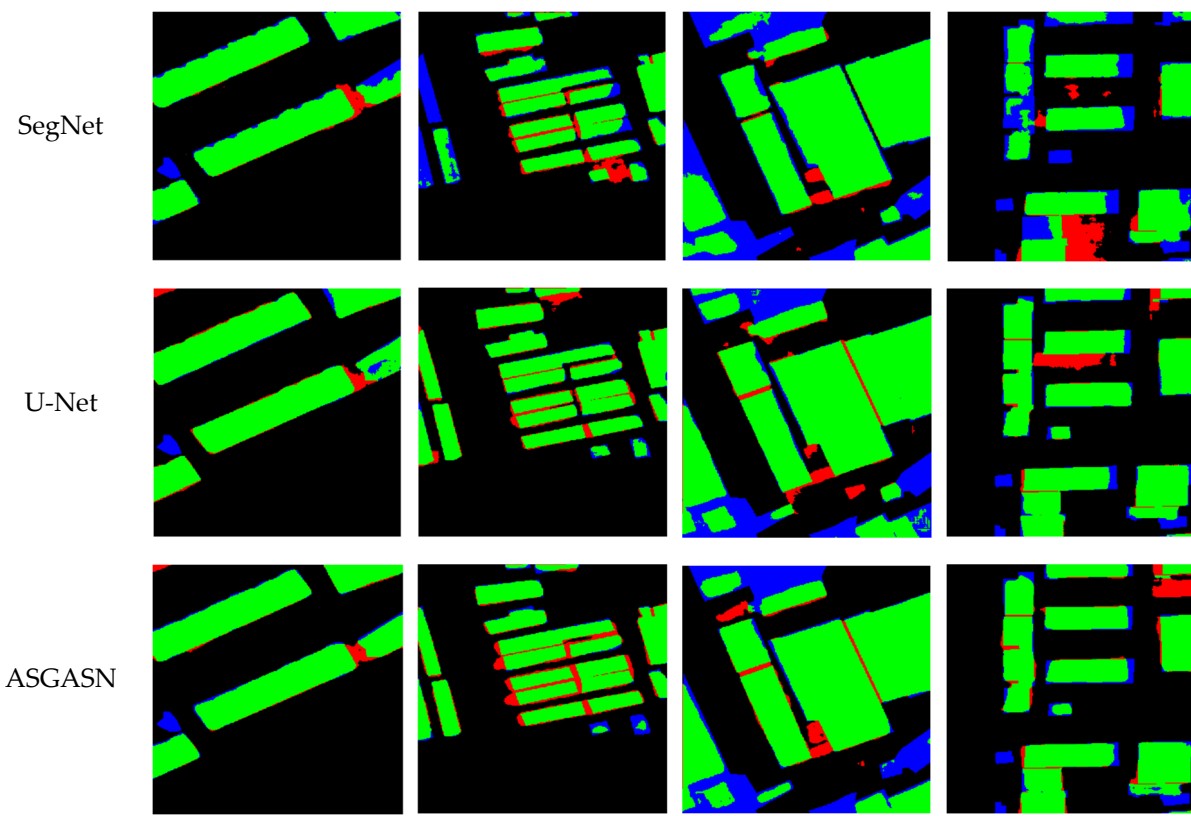

**Figure 12.** Examples of segmentation results by different models on the CHN dataset. The green, red, blue, and black pixels of the maps represent the true positive, false positive, false negative, and true negative predictions, respectively.

**Table 3.** Quantitative comparison with the state-of-the-art models on the CHN dataset. The highest value for each metric is marked as bold.

| Metrics | Methods | Image1 | Image2 | Image3 | Image4 | Mean |
|---|---|---|---|---|---|---|
| OA | FCN8s | 0.861 | 0.883 | 0.812 | 0.811 | 0.841 |
| | PSPNet | 0.917 | 0.883 | 0.781 | 0.813 | 0.848 |
| | SegNet | 0.967 | 0.932 | 0.818 | 0.882 | 0.899 |
| | U-Net | 0.961 | **0.961** | 0.846 | 0.935 | 0.925 |
| | ASGASN | **0.976** | 0.946 | **0.848** | **0.937** | **0.926** |
| Precision | FCN8s | **0.967** | 0.758 | 0.921 | **0.959** | 0.901 |
| | PSPNet | 0.946 | 0.882 | 0.964 | 0.941 | 0.933 |
| | SegNet | 0.958 | 0.866 | **0.972** | 0.943 | **0.934** |
| | U-Net | 0.949 | **0.883** | 0.963 | 0.932 | 0.931 |
| | ASGASN | 0.961 | 0.852 | 0.953 | 0.919 | 0.921 |
| Recall | FCN8s | 0.644 | 0.848 | 0.801 | 0.641 | 0.733 |
| | PSPNet | 0.741 | 0.695 | 0.744 | 0.674 | 0.713 |
| | SegNet | 0.887 | 0.821 | 0.719 | 0.729 | 0.789 |
| | U-Net | 0.875 | **0.948** | 0.813 | 0.896 | 0.833 |
| | ASGASN | **0.944** | **0.948** | **0.818** | **0.901** | **0.902** |
| F1-score | FCN8s | 0.773 | 0.801 | 0.856 | 0.768 | 0.799 |
| | PSPNet | 0.831 | 0.777 | 0.841 | 0.785 | 0.808 |
| | SegNet | 0.921 | 0.843 | 0.827 | 0.822 | 0.853 |
| | U-Net | 0.911 | **0.914** | 0.882 | **0.914** | 0.905 |
| | ASGASN | **0.952** | 0.897 | **0.883** | 0.910 | **0.911** |
| IoU | FCN8s | 0.631 | 0.668 | 0.749 | 0.624 | 0.688 |
| | PSPNet | 0.710 | 0.636 | 0.724 | 0.647 | 0.679 |
| | SegNet | 0.854 | 0.729 | 0.705 | 0.698 | 0.746 |
| | U-Net | 0.836 | **0.843** | 0.789 | **0.842** | 0.827 |
| | ASGASN | **0.908** | 0.814 | **0.791** | 0.834 | **0.836** |

## 5. Discussion

### 5.1. About the Proposed ASGASN Model

In recent years, there have been breakthroughs in deep learning architectures for semantic annotation of high-resolution images, which provide a completely new way of thinking for high-precision extraction of buildings. Some of these advanced FCN-based models (e.g., USPP [2], ARC-Net [4], and MC-FCN [47]) have improved the learning ability of image features and have better classification results. However, the contours extracted by existing networks are not accurate enough in the face of identical objects of various shapes, textures and sizes in high-resolution remote sensing images.

In this paper, we designed a new generative adversarial network with skip connections and ASPP for semantic segmentation, called ASGASN. The model has three key innovations. (1) The main contribution of this study is to apply the self-adversarial property of GAN to the automatic segmentation of buildings and develop a new network named ASGASN. While the more common segmentation networks are mainly based on the FCN architecture, which mainly controls the training of the network through the loss function and lacks the self-optimization of the network. We can intuitively see from the content of the red box in Figure 11 that the adversarial learning feature of ASGASN can extract some buildings that cannot be recognized by the FCN framework.(2) In order to better obtain the detailed features of the building outline, the segmentation network of the architecture uses a larger convolutional kernel and global convolutional block to expand the semantic information receiving domain, and the classical combination of ASPP+ skip connection is added to consider the contextual semantic information, which effectively improves the sensitivity of the model to the building. (3) To improve the discriminator network's ability to discriminate true or false, the multi-level downsampled feature maps are fused and ultimately guide the training of the entire ASGASN. Through these three innovations, ASGASN achieves a good performance improvement and can extract different types of building contours more accurately. The OA of the ASGASN on the WHU and China typical city building datasets are 97.4% and 92.6%, respectively, with F1-scores of 94.4% and 91.1%, respectively, and IoU values of 89.4% and 83.6%, respectively. Moreover, these results demonstrate the practical performance of ASGASN in real-world application scenarios.

### 5.2. Limitations

Despite the favorable performance in both qualitative analysis and quantitative evaluation, ASGASN still has certain limitations. The learning of the ASGASN model relies on a large number of training samples and performs poorly in classes with few training samples. In some areas, such as remotely sensed feature extraction, large amounts of data often lead to large costs. This requires our model to perform well against small sample data. In the future, it is critical to address small training samples through data enhancement techniques and semisupervised semantic segmentation techniques, with a focus on improving the accuracy of small sample categories.

## 6. Conclusions

In this paper, we propose a GAN framework called ASGASN for efficient and accurate automatic building segmentation from high-resolution remote sensing images. AS-GASN consists of a segmentation network and a discriminator network, which are used to obtain building segmentation results and to distinguish segmentation results from ground truth, respectively. Model training as well as comparative analysis experiments based on WHU dataset as well as a CHN dataset show that ASGASN extracts smoother and more accurate building contour edges, higher integrity of building interiors, and better performance for different types of buildings. The following conclusions are obtained based on the study.

(1) ASGASN using the adversarial training strategy can pay more attention to the relationship between pixels, improve the continuity of segmentation results, and make the extracted building boundaries clearer.

(2) ASGASN introduces depth-separable convolution and global convolution to im-prove the classification and localization accuracy of the model, and uses ASPP to improve the model's ability to perceive buildings at different scales. These measures allow the network to obtain building extraction results that are closer to the ground truth.

(3) The wide applicability of ASGASN for remote sensing images is greatly improved compared with other networks. The building extraction results on the WHU dataset show that ASGASN can get better extraction results for different types of buildings. Additionally, in the quantitative evaluation metrics of both datasets, the method in this paper achieves better score performance.

In the future, we plan on conducting research related to the field of deep neural network security such as robustness testing of models [48–50]. Meanwhile, we will consider combining with unsupervised classification and other algorithms to further improve the generalization ability of the model so that our model can be better applied to engineering fields such as urban planning and built-up area change detection.

**Author Contributions:** Conceptualization, W.Z. and M.Y.; methodology, W.Z.; software, W.Z.; val-idation, M.Y., W.Z. and X.C.; formal analysis, M.Y.; investigation, W.Z. and J.N.; resources, Y.L.; data curation, X.C.; writing—original draft preparation, M.Y.; writing—review and editing, W.Z.; visualization, W.Z.; supervision, M.Y.; project administration, M.Y.; funding acquisition, M.Y. All authors have read and agreed to the published version of the manuscript.

**Funding:** This research was financially supported by the National Natural Science Foundation of China (41801308) and the Open Fund Project of Key Laboratory of Earthquake Dynamics in Hebei Province (FZ212203).

**Institutional Review Board Statement:** Not applicable.

**Informed Consent Statement:** Not applicable.

**Data Availability Statement:** WHU dataset available at http://gpcv.whu.edu.cn/data/ (accessed on 21 April 2022); CHN dataset available at https://www.scidb.cn/en/detail?dataSetId=806674532768153600&dataSetType=journal (accessed on 21 April 2022).

**Acknowledgments:** The authors would like to thank the team from Wuhan University and China University of Geo-sciences (Wuhan) for providing the remote sensing dataset used in this study.

**Conflicts of Interest:** The authors declare no conflict of interest.

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
