# Peer review of "An End-to-End Atrous Spatial Pyramid Pooling and Skip-Connections Generative Adversarial Segmentation Network for Building Extraction from High-Resolution Aerial Images"

_applsci, doi:10.3390/app12105151_

Round 1

Reviewer 1 Report

The manuscript with the title An End-to-End Atrous Spatial Pyramid Pooling and Skip-Connections Generative Adversarial Segmentation Network for Building Extraction from High-Resolution Aerial Images after Minor Revision

In my opinion, the subject of this work is relevant for the Journal Applied Sciences after

approval of Minor revision.

The topic of the paper is very interesting and important in the connection between building analysis with the help of High-Resolution Aerial Images. The journal readers of MDPI Applied Sciences seek only quality papers. 

First, before all, the structure of the paper is divided into the next sections and sub-sections (i.e. Abstract, Introduction, Methods, Proposed Network ASGASN, Segmentation Network, Subsubsection Atrous Spatial Pyramid Pooling, Residual Block, Global Convolutional Block, Discriminator Network, Loss Function, Experiment Dataset and Evaluation, Experiment Data, Data Processing, Experiment settings, Evaluation Metrics, Model Comparisons, Results, Experimental Results on the WHU Dataset, Experimental Results on the CHN Dataset, Discussion, About the proposed ASGASN model, Limitations, Conclusions).

In the section of Abstract, it is necessary to add sentences which explained sub-pixel and pixel analysis.

The only concern within this manuscript is to add new Section or Sub-section in the section of Methods with the title Pixel or Sub-pixel analysis. In this recommended section it is necessary to put more sentences which explained pixel analysis of infrastructure and their application in some open-source software’s. For example, SAGA or QGIS, etc. In that way I recommend to the authors the next references. In these references authors can find practical applications of pixel and sub-pixel analysis.

-Valjarević, A., Filipović, D.,Valjarević, D., Milanović, M., Milošević, S., Živić, N., Lukić, T. (2020). GIS and remote sensing techniques for the estimation of dew volume in the Republic of Serbia. Meteorological Applications, 27(3): 1-14. DOI: 10.1002/met.1930.  

-Vikhamar, D., Solberg, R., 2003. Subpixel mapping of snow cover in forests by optical remote sensing, Remote Sensing of Environment, 84(1), 69-82. https://doi.org/10.1016/S0034-4257(02)00098-6.

The section Conclusion is overall short. This section must be immediately extending. In this section the authors must place sentences which explain the main goals of this research and potential application of presented methods.

This manuscript of course deserves to be published, after Minor revision. This manuscript has science potential and it describe rare subject and can be important for readership in the world.

In the end, I recommend Minor Revision.

Good luck to the authors

The Reviewer#1

Author Response

Point 1: In the section of Abstract, it is necessary to add sentences which explained sub-pixel and pixel analysis. 

Response : Thank you very much for your comments. We add the relevant analysis in the section of Abstract which are highlighted in red. Thanks!

Point 2: The only concern within this manuscript is to add new Section or Sub-section in the section of Methods with the title Pixel or Sub-pixel analysis. In this recommended section it is necessary to put more sentences which explained pixel analysis of infrastructure and their application in some open-source software’s. For example, SAGA or QGIS, etc. 

Response: Thank you very much for your comments. We add a new Sub-section of Pixel analysis in the section of Methods, in which pixel analysis and some of their applications in open-source software are described (lines 256-266). Thanks!

Point 3: The section Conclusion is overall short. This section must be immediately extending. In this section the authors must place sentences which explain the main goals of this research and potential application of presented methods.

Response: Thank you very much for your comments. We have expanded the section Conclusion to provide more detail on the main goals of this research and the potential application of proposed methods (lines 448-473). Thanks!

Reviewer 2 Report

This paper deals with an exciting topic. The article has been read carefully, and some minor issues have been highlighted in order to be considered by the author(s).

#1 What is the motivation of this paper?

#2 What is the contribution and novelty of this paper?

#3 What is the advantage of this survey paper?

#4 Which evaluation metrics did you used for comparison?

#5 It would be good if security domains for the deep neural network would be reflected in the related work such as BlindNet backdoor: Attack on deep neural network using blind watermark, Medicalguard: U-net model robust against adversarially perturbed images, Data Correction For Enhancing Classification Accuracy By Unknown Deep Neural Network Classifiers.

#6 In Figure 1, what size is each layer?

Round 2

Reviewer 2 Report

I recommend the acceptance.